# Potential Mechanism of Action of Current Point-of-Care Autologous Therapy Treatments for Osteoarthritis of the Knee—A Narrative Review

**DOI:** 10.3390/ijms22052726

**Published:** 2021-03-08

**Authors:** Jennifer Woodell-May, Kathleen Steckbeck, William King

**Affiliations:** 1Zimmer Biomet, 56 East Bell Drive, Warsaw, IN 46580, USA; kathleen.steckbeck@zimmerbiomet.com; 2Owl Manor, 720 East Winona Avenue, Warsaw, IN 46580, USA; william.king@owlmanormedical.com

**Keywords:** platelet-rich plasma, autologous anti-inflammatory, concentrated bone marrow aspirate, osteoarthritis, intra-articular injection, mechanism of action

## Abstract

Osteoarthritis (OA) is a progressive degenerative disease that manifests as pain and inflammation and often results in total joint replacement. There is significant interest in understanding how intra-articular injections made from autologous blood or bone marrow could alleviate symptoms and potentially intervene in the progression of the disease. There is in vitro an in vivo evidence that suggests that these therapies, including platelet-rich plasma (PRP), autologous anti-inflammatories (AAIs), and concentrated bone marrow aspirate (cBMA), can interrupt cartilage matrix degradation driven by pro-inflammatory cytokines. This review analyzes the evidence for and against inclusion of white blood cells, the potential role of platelets, and the less studied potential role of blood plasma when combining these components to create an autologous point-of-care therapy to treat OA. There has been significant focus on the differences between the various autologous therapies. However, evidence suggests that there may be more in common between groups and perhaps we should be thinking of these therapies on a spectrum of the same technology, each providing significant levels of anti-inflammatory cytokines that can be antagonists against the inflammatory cytokines driving OA symptoms and progression. While clinical data have demonstrated symptom alleviation, more studies will need to be conducted to determine whether these preclinical disease-modifying findings translate into clinical practice.

## 1. Introduction

Osteoarthritis (OA) is a degenerative and disabling articulating joint disease that affects both younger, more active patients (e.g., patients with trauma or who have prolonged participation in highly demanding sports) and the elderly [1,2]. The disease is progressive and debilitating, eventually resulting in pain that may be so severe that restive sleep is impossible, along with life-altering loss of function. 

Surgical intervention is clinically successful, and widely used, in treating severe degenerative OA; however, treatment modalities for less advanced OA are associated with varying rates of success. Current treatment options include non-steroidal anti-inflammatory drugs (NSAIDs), corticosteroid injections, and hyaluronic acid (HA) injections. Although these treatments can relieve pain temporarily for some OA patients, they may not address the biological mechanisms causing the disease [3].

Although OA is classified as a non-inflammatory disease, inflammation is implicated in many symptoms and in OA progression. Pro-inflammatory cytokines involved in OA development include interleukin-1 (IL-1), tumor necrosis factor alpha (TNFα), interleukin-6 (IL-6), and interleukin-8 (IL-8) [4]. The cytokines associated with inflammation in OA, primarily interleukin-1 beta (IL-1β) and TNFα, are also implicated in cartilage matrix breakdown [5]. These cytokines induce chondrocytes to produce matrix metalloproteinases (MMPs) that in turn are responsible for cartilage matrix degradation [6]. These cartilage matrix breakdown products in synovial fluid are thought to increase synovial inflammation [7], creating a positive feed-forward loop that increases inflammation and cartilage breakdown (Figure 1).

As IL-1β and TNFα play important roles in inflammation and cartilage breakdown, inhibition of these cytokines may limit inflammation and matrix degradation. Consequently, inhibition of these proteins may constitute an effective OA therapy. The anti-inflammatory cytokine interleukin-1 receptor antagonist (IL-1ra), a competitive IL-1 receptor antagonist, blocks the signaling activity of IL-1 while exhibiting no signal-inducing activity itself [8]. Soluble forms of the IL-1 receptor (sIL-1R) can also bind with IL-1, reducing IL-1 biologic activity by preventing it from binding to surface receptors on the cells [9]. Moreover, soluble forms of the cell receptors for TNFα, known as sTNF-RI and sTNF-RII, can bind to TNFα, preventing TNFα surface receptor binding and thus inhibiting cell signaling [10].

Over the past two decades, preparations from autologous blood or bone marrow to create platelet-rich plasma (PRP), autologous anti-inflammatories (AAIs), and concentrated bone marrow (cBMA or BMAC) have been studied in order to treat osteoarthritis. Each of these formulations are created from similar starting materials by combining plasma, platelets, white blood cells (including early lineages) in various combinations. Each formulation is in on a spectrum of similar mechanistic approaches to treating OA. Regulatory approval to treat OA vary for these formulations depending on the region in the world. The purpose of this review is to summarize the data that exist for each of these classes of autologous therapies and how each may intervene in the OA disease process.

## 2. Methods

Search terms in both PubMed database and Google Scholar included “platelet-rich plasma”, PRP, osteoarthritis, “cell culture”, preclinical, and “mechanism of action” in various combinations. The same combinations were repeated for “concentrated bone marrow aspirate”, “cBMA”, “BMAC”, “autologous anti-inflammatory”, and “AAI”. Additionally, the authors’ own knowledge and work in the field was included when relevant. A focus was made on the specific role that these therapies play in the treatment of OA, and therefore cartilage-specific studies were only included to expand on the potential role that cartilage repair has during treatment of OA.

## 3. Results

### 3.1. Platelet-Rich Plasma

Platelet-rich plasma (PRP), or platelet concentrate, is defined as an autologous therapy prepared point of care that contains a concentration of platelets higher than whole blood. This review will focus on PRP prepared by centrifugation that takes advantage of the density differences between the elements in peripheral blood allowing for separation of components. In general, PRP can be divided broadly into two classifications. The first preparation class is created with “slow” centrifugation speeds (e.g., 1500 rpm for 5 min) and have platelet concentrations 2–3 times above baseline collected in blood plasma (platelet-poor plasma; PPP), and with very few red blood cells or white blood cells. These PRPs are yellow in color. The second classification is prepared from “hard” spin cycles (e.g., 3200 rpm for 15 min) resulting in PRPs with platelet concentrations 5–9 above whole blood and typically with white blood cells also above baseline levels. This method typically contains some RBC and are red in color. One popular classification method to discern these two PRPs has focused on the differences in white blood cell concentrations, calling the products with low white blood cell concentrations leukocyte-poor platelet-rich plasma (LP-PRP) and those with higher WBC concentrations as leucocyte-rich platelet-rich plasma (LR-PRP). An evaluation of blood components can be compared by complete blood counts (CBC) (Table 1). Platelet concentrates can also be activated to form a fibrin clot, often referred to as platelet-rich fibrin (PRF). These gel products are often utilized in direct application to a surgical or wound site [11], but are not included in this OA review.

As PRP first became popular in orthopedics, clinical use was focused on both hard (bone) and soft tissues (tendon, muscle, etc.) because the platelets concentrated contain growth factors (PDGF, VEGF, TGF-β1, EGF) that are chemoattractive, proliferative, and angiogenic, therefore, playing a key role in signaling the wound healing cascade. When these mechanisms were translated into treating OA, the initial belief was that the growth factors in platelets would stimulate chondrocyte cell growth in the cartilage, thereby healing cartilage defects. There is ample evidence that PRPs can induce chondrocyte proliferation [15,16] in cell culture. However, chondrocyte proliferation effects have not yet translated into a clinical benefit when treating OA. PRP OA mechanism has evolved to include intervening in the inflammatory process. Interestingly, the factors that most likely contribute anti-inflammatory cytokines in PRP are found in the blood plasma and white blood cells and not in the platelets (Figure 2).

PRP in vitro mechanistic data in OA have focused on the differences between LP-PRP and LR-PRP. The cytokines found in a LP-PRP and LR-PRP reflect their platelet and white blood cell concentration (Figure 1). LP-PRP has lower concentrations of platelet-derived growth factors (PDGF, TGF- β1) than LR-PRP due to lower platelet concentrations (Table 1). LR-PRP has more white blood cell-derived cytokines over LP-PRP that include both pro-inflammatory cytokines (IL-1β, MMP-9) [17] and anti-inflammatory cytokines (IL-1ra) [12]. IL-1ra is produced from monocytes [18] and neutrophils [19]. Therefore, in order for a PRP to have high concentrations of IL-1ra, WBCs need to be included. In order to ensure that the PRP is not pro-inflammatory, the ratio of IL-1β and IL-1ra is important. In a common LR-PRP, the amount of IL-1β is 3.5 ± 1.0 pg/mL while the IL-1ra is 22,395 ± 12,900 pg/mL, giving a ratio between the two of 6369 ± 2321 [12]. In order to evaluate how much IL-1ra is required to block the activity of IL-1β, investigators intravenously injected IL-1β to induce fever in rabbits and then increasing concentrations of the antagonist IL-1ra to determine how much in excess was required. Symptoms were blocked 50% when IL-1ra was injected at 100-fold increase over IL-1β (100 IL-1ra:IL-1β) and were completely blocked by 1000-fold excess of IL-1ra (1000 IL-1ra:IL-1β) [20].

In addition to production of pro-inflammatory cytokines, in cell culture LR-PRP has demonstrated higher levels of synoviocyte cell death and pro-inflammatory cytokine production compared to LP-PRP or PPP [21]. As LR-PRP has higher concentrations of RBC, it has been observed that layers of RBC deposit on the top of a monolayer cell culture, inhibiting nutrient transfer, and could lead to increased cell death and stimulation of pro-inflammatory cytokines. Cell culture techniques with autologous therapies that contain higher red blood cell content can be improved by either suspending the treatment above the monolayer culture with a cell culture insert (Nunc™ Polycarbonate Membrane Inserts, Thermo Scientific™) or by creating a thrombin-activated releasate. Using these techniques, both LP-PRP and LR-PRP can stimulate chondrocyte cell proliferation [15,16]. Additionally, human intra-articular injections of LR-PRP to treat OA has not resulted in increased pro-inflammatory cytokines in synovial fluid or blood plasma [22]. 

A less lauded component of PRP that contributes anti-inflammatory cytokines is the platelet-poor plasma. All autologous therapies are collected in some volume of blood plasma (Figure 2). Blood plasma contains sIL-1R1, sTNF-RI, sTNF-RII, and A2M [12,23,24]. Interestingly, no matter what autologous therapy, these factors contributed from blood plasma will be present at least in concentrations similar to whole blood, and in some formulations increased levels over whole blood. When examining the anti-inflammatory mechanism of action in OA, the PPP contributes more anti-inflammatory cytokines than platelets. This is a novel approach to understanding the activity of a LP-PRP product in the treatment of OA. 

The in vitro positive effects on chondrocytes has translated into demonstrated benefit in animal models of chondral defects [25,26]. PRP has also demonstrated improved histology scores compared to saline in collagenase-induced OA rabbit model [27] and ACL transection rabbit model [28]. As treatment of OA is often focused on the alleviation of pain, in a randomized control trial in dogs with naturally occurring OA, PRP treated dogs had significant decrease in lameness grades compared to saline control subjects [29]. 

While there has been evidence of intra-articular injections of both LR-PRP and LP-PRP [30,31] providing clinical benefits for patients with knee OA pain, to date, there have been no correlation between any component of the PRP to clinical outcomes. The question still remains whether platelets, white blood cells, or, in fact, the blood plasma, contains the necessary ingredients to treat knee OA pain or to have any disease-modifying effect. 

### 3.2. Autologous Anti-Inflammatory

Autologous anti-inflammatories (AAIs) are autologous blood-derived technologies that focus on concentrating high levels of anti-inflammatory cytokines [32,33]. As the focus on platelet-rich plasma is to concentrate platelets, the AAI designation was created to differentiate from products focused on platelets to those focused on the concentration of anti-inflammatory cytokines. As mentioned earlier, IL-1ra is produced from monocytes and [18] and neutrophils [19]. IL-1ra correlates to the number of WBCs contained in the output [34]. Therefore, AAIs typically utilize strategies to maximize WBC or to collect cytokines produced from WBCs.

Two examples of AAIs are autologous protein solution (APS) and autologous conditioned serum (ACS) [12]. APS is processed through a two-step centrifugation method where anticoagulated peripheral blood is centrifuged to isolate a leukocyte-rich buffy coat suspended in plasma. The buffy coat is mixed with polyacrylamide beads to absorb water; the output is then centrifuged a second time creating a WBC concentrate output suspended in a concentrated PPP [33,35]. APS is designed as a point-of-care single injection OA therapy [36,37]. ACS is produced by incubating venous blood at 37 °C in borosilicate glass-bead containing tubes for 6–24 h. The blood clots and the WBCs expel IL-1ra and other cytokines. Following incubation, tubes are centrifuged and serum is removed, aliquoted into syringes, and stored frozen −20 °C for future multiple injections [32]. The major difference between APS and ACS is that APS output contains WBCs and ACS is cell-free serum collected from incubated WBCs.

The composition of AAIs is designed to inhibit the synergistic feed-forward progression of IL-1β and TNF-α by concentrating their respective antagonists (IL-1ra, sIL-1RII, sTNF-RI, and sTNF-RII) as well as multiple other cytokines and growth factors [32,38] with the goal of intervening in the pro-inflammatory cytokines (IL-1β and TNFα) that cause the degradation of OA [39] (Table 1).

Since AAIs have been specifically created with OA treatment in mind, there is an extensive amount of preclinical data supporting their mechanism of action of intervening in the IL-β and TNFα catabolic pathways described earlier. For example, APS has been shown to reduce MMP-13 production from IL-1β and TNFα-stimulated chondrocytes [35] and decreased IL-8 production from IL-1β-stimulated activated macrophages [40]. AAIs have also demonstrated ability to reduce matrix degradation. In a stimulated equine cartilage and synovium explant co-culture, AAIs significantly downregulated IL-1β expression in cartilage, reduced catabolic cartilage production of PGE2, and downregulated MMP-1 in the synovium [41]. AAIs also upregulated type II collagen and aggrecan expression [41]. AAIs provided chondroprotective effects and decreased matrix degradation in cartilage explants treated with IL-1 and TNFα [38,41]. 

When translating cell culture experiments into animal models, an intra-articular injection of APS demonstrated cartilage matrix protection by improved cartilage histology scoring compared to animals treated with saline in both a meniscal-tear OA model [42] and IL-1β-induced OA model [43] in athymic rats. Single injections of APS have also demonstrated significant improvement in lameness compared to a single injection of saline in both equine [44] and canine [45] studies.

AAIs have demonstrated early clinical evidence linking cytokine and cellular content and clinical response in patients with OA. Wasai et al. found that the IL-1ra concentration in APS positively correlated with changes in the clinical outcome scores in subjects with knee OA [46]. In a clinical study of OA subjects treated with a single injection of APS, characterization analysis showed 85.7 % of subject’s APS had an IL-1ra:IL-1 ratio greater than 1000 or a WBC count greater than 30 k/μL. These subjects were also high OMERACT-OARSI clinical responders six months post-injection (Figure 3) [34,47]. Interestingly, this is the same ratio (1000 IL-1ra:IL-1β) that demonstrated complete blockage of an IL-1β fever-induction model in rabbits [20]. These early results suggest that WBC, and consequently IL-1ra concentrations, can improve outcomes in OA subjects.

Multiple clinical trials in the management of knee OA have been completed using AAIs as they have proven capable of producing high concentrations of anti-inflammatory cytokines from healthy subjects as well as subjects diagnosed with OA [32,33,46] Angadi et al. has recently reviewed and compared the current landscape of AAI clinical evidence and concluded their safety profiles satisfactory for clinical use and presented similar risk profiles of general intra-articular injections [48]. Two knee OA randomized controlled trials of ACS intra-articular injections, Yang et al. and Baltzer et al., demonstrated OA symptom improvements of ACS patients over placebo injections [49,50]. Several studies of a single injection of APS in knee OA have also demonstrated clinical benefits [36,37,51,52,53].

### 3.3. Concentrated Bone Marrow Aspirate 

Devices that concentrate bone marrow aspirate have been explored to address osteoarthritis in preclinical models [54,55,56,57,58] and clinical trials [59,60,61,62]. Understanding what is in cBMA could inform how it might address osteoarthritis. The first proposed MOA for cBMA in treating OA was attributed to “stem cells [63].” However, cBMA contains a low number of stem cells in comparison to culture-expanded stem cell approaches [64]. An emerging understanding of cBMA is that it is a WBC, progenitor cell, and platelet-rich product which enables it to contain a high concentration of anti-inflammatory cytokines and anabolic growth factors [65]. Understanding that cBMA is in the same spectrum of options with PRPs and AAIs will help scientists and clinicians better understand the potential of this option. 

The output of cBMA devices is determined by the components of bone marrow, the surgeon’s aspiration technique, and the devices used to concentrate the cells and platelets. Bone marrow is a complex tissue made of hematopoietic stem cells (HSCs) which form blood [66], mesenchymal stem cells (MSCs) which form connective tissues [67], endothelial progenitor cells (EPCs) which form blood vessels [68], WBCs of mixed phenotypes, and platelets. Ultimately, the function of bone marrow is to create new blood cells. Stem cells are rare in bone marrow: Caplan has estimated that 1:10,000 cells in bone marrow is an MSC in newborns and that percentage declines with age [69]. It is important to note that the same white blood cells and platelets that are in bone marrow migrate to the blood stream. They are connected and part of the same system [70]. A surgeon’s bone marrow aspiration technique impacts what cells are obtained for processing. It is not possible to obtain pure bone marrow without some peripheral blood dilution. Hernigou and colleagues have shown that smaller volume bone marrow draws, smaller volume syringes, and changing aspirate locations can minimize the dilution of bone marrow cells with peripheral blood [71]. Furthermore, different locations from aspiration could vary the cellular content of bone marrow aspirate [72]. While these techniques have been shown to be “best practices”, these approaches are considered by some surgeons to be not pragmatic. Therefore, it is likely that many surgeons may be delivering a product that is closer to a blood-derived PRP than cBMA. 

Providing bone marrow stem cells (BMSCs) have been one of the rationales proposed as for the intra-articular injection of cBMA for OA. Cultured BMSCs have had beneficial effects in cell-culture models of osteoarthritis including their differentiation into chondrocytes [73] and inhibiting inflammation [74]. BMSCs have had positive effects in a large animal model of OA that was used in support of a larger human program. Specifically, culture-expanded BMSCs stimulated the regeneration of meniscal tissue and slowed the progression of OA in a medial meniscus excision/anterior cruciate resection goat model [75]. However, these large animal positive results did not translate into regulatory approvals in humans. In a Phase I/II clinical study, 55 patients who underwent a partial medial meniscectomy received two doses of allogeneic mesenchymal stem cells and a vehicle control. While the cells showed safety and there was some new meniscal tissue evident in a portion of subjects twelve months post-meniscectomy [76], the study sponsor did not proceed to a confirmatory trial. Currently, there are no approved cultured stem cell therapies for treatment of osteoarthritis in the United States. Potential gaps in translation of these stem cells include a lack of clear phenotypic parameters for the cells, no set mechanism of action to optimize, and a complex manufacturing process [77]. Together, these data and history suggest that if cBMA has a role in addressing OA, then other non-stem cell components must play a significant role. 

There has been significant characterization work performed on cBMA. cBMA contains a high concentration of WBC and therefore has a high concentration of IL-1ra [78]. Indeed, there is a strong and significant correlation between the WBC concentration in cBMA with IL-1ra (R^2^ = 0.92) [14]. In addition to IL-1ra, cBMA contains high concentrations of other anti-inflammatory cytokines like sIL-1RII, sTNF-RI, and sTNF-RII and low concentrations of inflammatory cytokines like IL-1β and TNFα (Table 1). Interestingly, cBMA is the only autologous therapy where the ratio of IL-1ra:IL-1β decreases in the concentrated version over the starting material (Table 1). While still very low, this is due to an increase in IL-1β by almost 4-fold (3.0 ± 1.1 pg/mL in BMA and 14.5 ± 11.4 pg/mL in cBMA). However, cBMA has by far the most IL-1ra concentration of any autologous blood product with an average of 73,978± 39,464 pg/mL. 

Just as with PRP, cBMA has significant concentrations of anabolic and angiogenic growth factors such as TGF-β1, PDGF-AB, PDGF-BB, and EGF [14]. These trophic factors are also secreted from cultured mesenchymal stem cells in culture through extracellular vesicles (EVs). EVs could be a method to utilize the signal factors from allogenic cell sources such as umbilical cord or Wharton’s jelly. EVs from MSCs have been shown to be chondroprotective and anti-inflammatory in cell culture, improve histology scores in both RA and OA animal models, and demonstrates early promise in Phase I/II clinical studies [79].

In addition to IL-1ra, the WBC concentration in cBMA is also significantly correlated with colony forming units–fibroblasts (CFU-F) [80], which is a surrogate marker for BMSCs [81]. cBMA has other progenitor cells including concentrated HSCs and EPCs [82], whose intra-articular role in osteoarthritis has not been extensively characterized. These separate anti-inflammatory and pro-angiogenic properties of cBMA could have differential effects when its injected in different locations. 

cBMA is delivered to an OA joint via intra-articular injection. However, cBMA has been used in cartilage repair techniques in combination with varying biomaterials and drug delivery systems including hydrogels and microspheres, and in combination with microfracture surgery. These delivery techniques are intended to increase their residence time and bioavailability of the bioactive factors released from the cells [83]. These approaches have produced durable pain relief in several single-arm studies [84,85,86]. The intra-articular injection of cBMA for treatment of OA has produced mixed results in randomized and controlled trials. For example, in one study 25 patients with bilateral knee pain received injections of saline in one knee and cBMA in the contralateral knee. Clinical improvements were seen in both knees, but there wasn’t a statistical difference between saline and cBMA [59]. In another study, 90 patients with symptomatic knee OA received intra-articular injection of either PRP or cBMA. Both groups had clinically improved symptoms 12 months post-injection, but there were not significant differences between groups. To date, clinical evidence has demonstrated relief of OA symptoms but has not demonstrated reliably that they regrow lost cartilage tissue in OA. 

A recent model of OA has been put forward which conceptualizes the “joint as an organ.” Subchondral bone plays an important role in OA. Indeed, changes in subchondral bone can alter the function and pain of the whole joint [87,88,89]. Bone marrow lesions are hypothesized to originate from a traumatic event and have been shown to correlate with pain [90,91,92]. The proposed mechanism of action for cBMA in these lesions could be its angiogenic molecules and cells recruiting new blood vessels and bone turnover in the lesion, restoring bone health [93]. In a canine subchondral bone lesion model, cBMA and calcium phosphate-injected lesions enhanced knee range of motion and limb loading through improved trabecular bone remodeling [94]. Direct injection of cBMA into an avascular necrosis defect along with decompression has also demonstrated some promise [95]. This general approach of delivering cBMA with or without a carrier has been explored in clinical case series [96,97,98]. However, further clinical evidence would likely be required. 

## 4. Conclusions

There is significant interest in autologous therapies to treat OA patients, both to alleviate symptoms and to potentially delay progression of the disease. While there has been a focus on the differences between these therapies, particularly the presence or absences of white blood cells, the cell culture and animal studies suggest more in common than not. As we examine the mechanistic evidence of each of the therapies, they all have in common the production of anti-inflammatory cytokines at varying concentrations that can be shown both in cell culture and in animal models to delay disease progression. If autologous therapies’ mechanism of action is via modifying the local inflammatory environment in the joint, and PRP, AAIs, and cBMA have significant concentrations of anti-inflammatory cytokines, this could explain why all three therapies have demonstrated some level of clinical efficacy. It may be true that the proliferative cytokines derived from platelets, and found in all three therapies, may also play a role in OA, though beneficial effects have been demonstrated in preclinical models but not yet in a clinical setting. 

To date, clinical evidence suggests that they can all alleviate symptoms from patients with OA, but have not yet definitively demonstrated disease modification. Correlations between autologous therapy content and clinical outcomes have been the holy grail in autologous therapies. There is early evidence that potentially the IL-1ra:IL-1b ratio in the output of the therapy may play a role. This fact challenges the dogma that white blood cells, whether sourced from peripheral blood or from bone marrow, should be eliminated from autologous therapies altogether in the treatment of OA. Further studies will be needed to confirm these hypotheses explored in this review. Ultimately, convincing clinical evidence of disease modification will require long-term studies that utilize imaging or other surrogate markers including biomarkers or delay of total joint replacement. 

## Figures and Tables

**Figure 1 ijms-22-02726-f001:**
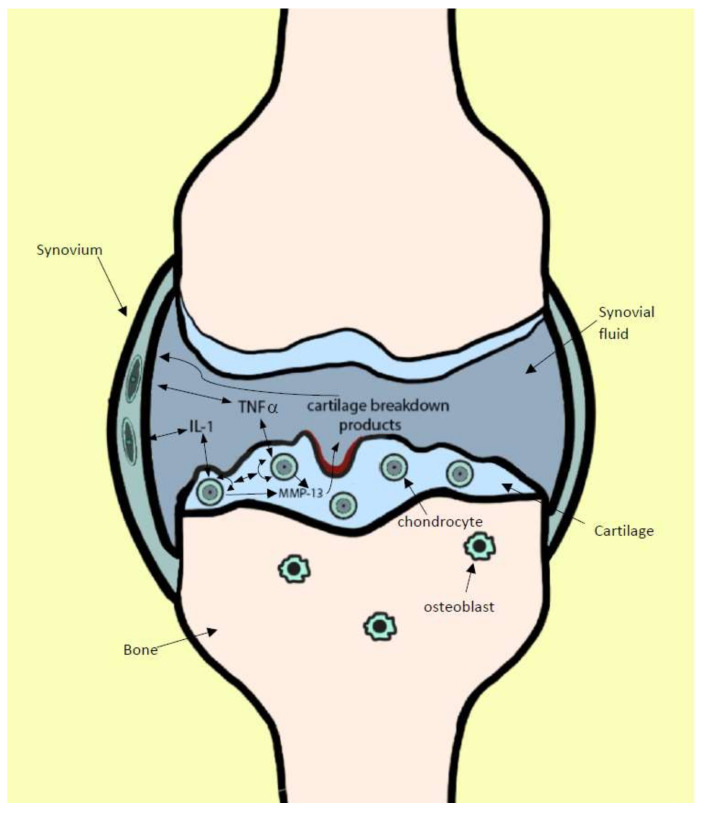
Model of feed-forward loop of pro-inflammatory cytokine-driven OA progression. IL-1β and TNFα bind to chondrocytes and induce expression of MMP-13. MMP-13 causes cartilage matrix breakdown. Cartilage breakdown products inflame cells in the synovium, inducing more production of IL-1β and TNFα.

**Figure 2 ijms-22-02726-f002:**
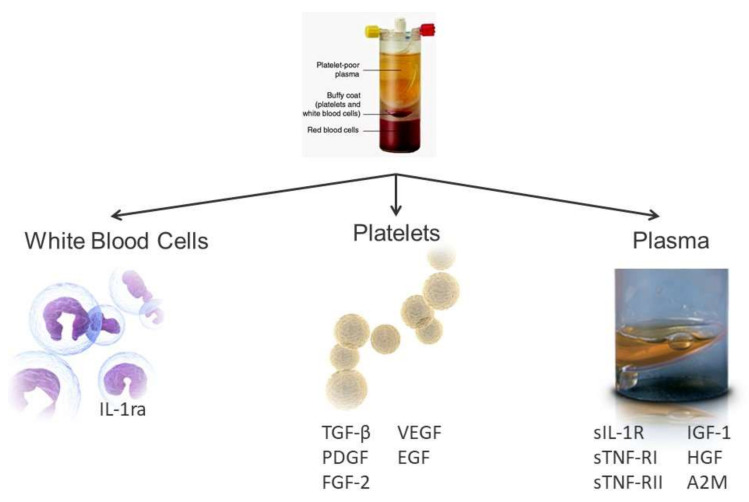
Cytokine contribution from components of blood used to make autologous therapies.

**Figure 3 ijms-22-02726-f003:**
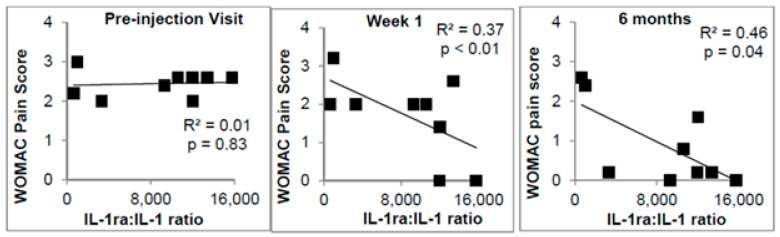
Correlation of WOMAC pain score and ratio of IL-1ra:IL-1β injection in OA subjects [34,47].

**Table 1 ijms-22-02726-t001:** Complete blood counts (CBC) of common autologous therapies. (PPP: platelet-poor plasma; LP-PRP:leucocyte-poor platelet-rich plasma; LR-PRP: leucocyte-rich platelet-rich plasma; APS: autologous protein solution; ACS: autologous conditioned serum; BMA: bone marrow aspirate; cBMA: concentrated bone marrow aspirate; BL: below lower limit; NC: not calculated, NM: not measured).

	WBC (k/ul)	PLT (k/ul)	RBC (M/ul)	IL-1ra (pg/mL)	sIL-1RII (pg/mL)	sTNF-RII (pg/mL)	IL-1β (pg/mL)	IL-1ra:IL-1 ratio
Whole blood [12]	5.4 ± 1.8	175 ± 70	5.5 ± 1.1	5665 ± 2318	7135 ± 1766	1125 ± 253	3.4 ± 2.0	4842 ± 2756
PPP [13]	0.1 ±0.0	28 ± 9.3	0.00 ± 0.00	296 ± 141	19,922 ± 2938	3080 ± 635	BL	NC
LP-PRP [13]	1.5 ± 2.0	399 ± 108	0.04 ± 0.06	673 ± 741	15,596 ± 2159	2894 ± 689	BL	NC
LR-PRP [12]	28.1 ± 6.9	1745 ± 439	0.9 ± 0.3	22,395 ± 12,900	NM	NM	3.5 ± 1.0	6369 ± 2321
APS [12]	46.5 ± 14.0	707 ± 444	1.5 ± 1.1	30,853 ± 16,734	20,483 ± 5819	9492 ± 1387	3.8 ± 0.8	8535 ± 3999
ACS [12]	0.0 ± 0.0	14 ± 6	0.0 ± 0.0	1618 ± 675	15,678 ± 2356	2696 ± 679	14.7 ± 14.8	291 ± 256
BMA [14]	22 ± 10	116 ± 30	4.1 ± 0.3	18,110 ± 6681	6768 ± 1995	1292 ± 153	3.0 ± 1.1	6154 ± 1357
cBMA [14]	133 ± 91	885 ± 201	1.3 ± 0.2	73,978 ± 39,464	9814 ± 3199	3932 ± 1301	14.5 ± 11.4	5856 ± 2745

## Data Availability

The data presented in this study are available on request from the corresponding author.

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
