# Peer review of "Potential Mechanism of Action of Current Point-of-Care Autologous Therapy Treatments for Osteoarthritis of the Knee—A Narrative Review"

_ijms, 2021, doi:10.3390/ijms22052726_

Round 1

Reviewer 1 Report

Woodell-May and coworkers summarized the potential mechanism of action of current point-of-care autologous therapy treatments for osteoarthritis of the knee. It is a well written mini-review. I recommend it for publication in IJMS after the following points are addressed.

  1. The resolution of the figures should be higher.
  2. Please add the copyright information into the caption of figures if the figures are reproduced from the literature.
  3. It will be better if the authors could add some perspective at the end of the Conclusion section.
  4. Several recent reviews (doi.org/10.3390/cells9061343; doi.org/10.1016/j.actbio.2019.01.061) related to this topic should be included.
  5. For the readers, it is better to add one or two figures, for example, a schematic of different techniques using concentrated bone marrow aspirate.

Author Response

Thank you for your careful review and consideration of our manuscript.  We hope we have addressed your concerns adequately. 

Reviewer 1

  1. The resolution of the figures should be higher.

We have increased the resolution of the figures.

  1. Please add the copyright information into the caption of figures if the figures are reproduced from the literature.

The figures are all original and not previously copyrighted. 

  1. It will be better if the authors could add some perspective at the end of the Conclusion section.

Thank you for this suggestion.  The following was added to the conclusion section. (Lines 336-350).

If autologous therapies’ mechanism of action is via modifying the local inflammatory environment in the joint, and PRP, AAI, and cBMA have significant concentrations of anti-inflammatory cytokines, this could explain why all three therapies have demonstrated some level of clinical efficacy. It may be true that the proliferative cytokines derived from platelets, and found in all three therapies, may also play a role in OA, though beneficial effects have been demonstrated in preclinical models but not yet in a clinical setting.

To date, clinical evidence suggests they can all alleviate symptoms from patients with OA, but have not yet definitely demonstrated disease-modification. Correlation between autologous therapy content and clinical outcomes have been the holy grail in autologous therapies. There is early evidence that potentially IL-1ra:IL-1b ratio in the output of the therapy may play a role. This fact challenges the dogma that white blood cells, whether sourced from peripheral blood or from bone marrow, should be eliminated from autologous therapies altogether in the treatment of OA. Further studies will be needed to confirm these hypotheses explored in this review. However, convincing clinical evidence of disease-modification will require long term studies that utilize imaging or other surrogate markers including biomarkers or delay of total joint replacement.

  1. Several recent reviews (doi.org/10.3390/cells9061343; doi.org/10.1016/j.actbio.2019.01.061) related to this topic should be included.

Thank you for these references.  Both were added to the manuscript.

Lines 291- 297

Just as with PRP, cBMA has significant concentrations of anabolic and angiogenic growth factors such as TGF-β1, PDGF-AB, PDGF-BB, and EGF [1]. These trophic factors are also secreted from cultured mesenchymal stem cells in culture through extracellular vesicles (EVs). EVs could be a method to utilize the signal factors from allogenic cell sources such as umbilical cord or Wharton’s jelly. EVs from MSCs have been shown to be chondroprotective and anti-inflammatory in cell culture, improve histology scores in both RA and OA animal models, and demonstrates early promise in Phase I/II clinical studies [2].

Lines 303- 307

cBMA is delivered to an OA joint via intra-articular injection. However, cBMA has been used in cartilage repair techniques in combination with varying biomaterials and drug delivery systems including hydrogels and microspheres, and in combination with microfracture surgery. These delivery techniques are intended to increase their residence time and bioavailability of the bioactive factors released from the cells [3].

  1. For the readers, it is better to add one or two figures, for example, a schematic of different techniques using concentrated bone marrow aspirate

The reference provided above (Patel et al, 2019) gives a very good schematic of potential delivery techniques.  As we do not have the copyright for these images, I have referred to this paper and described in words the techniques. (Lines 289-293). An additional reference was added expanding on the injection of cBMA into bone (Martin at el, 2013). Lines 320-322.

Direct injection of cBMA into a avascular necrosis defect along with decompression has also demonstrated some promise [4]. This general approach of delivering cBMA with or without a carrier has been explored in clinical case series [5-7], however, further clinical evidence would likely need be required.

King, W., et al. Anti-Inflammatory Properties of the Output of an Autologous Bone Marrow Concentrating Device. in Orthopaedic Research Society. 2016. Orlando, FL.

  1. Arrigoni, C., et al., Umbilical Cord MSCs and Their Secretome in the Therapy of Arthritic Diseases: A Research and Industrial Perspective. Cells, 2020. 9(6).
  2. Patel, J.M., et al., Bioactive factors for cartilage repair and regeneration: Improving delivery, retention, and activity. Acta Biomater, 2019. 93: p. 222-238.
  3. Martin, J.R., M.T. Houdek, and R.J. Sierra, Use of concentrated bone marrow aspirate and platelet rich plasma during minimally invasive decompression of the femoral head in the treatment of osteonecrosis. Croatian Medical Journal, 2013. 54(3): p. 219-224.
  4. Kasik, C.S., et al., Short-Term Outcomes for the Biologic Treatment of Bone Marrow Edema of the Knee Using Bone Marrow Aspirate Concentrate and Injectable Demineralized Bone Matrix. Arthroscopy, Sports Medicine, and Rehabilitation, 2019. 1(1): p. e7-e14.
  5. Ankem, H.K., et al., Arthroscopic-Assisted Intraosseous Bioplasty of the Acetabulum. Arthroscopy techniques, 2020. 9(10): p. e1531-e1539.
  6. Hood JR, C.R. and J.R. Miller, The triad of osteobiology–Rehydrating calcium phosphate with bone marrow aspirate concentrate for the treatment of bone marrow lesions. Bone, 2016. 5(7): p. 8.

Reviewer 2 Report

The manuscript submitted to IJMS entitled “Potential mechanism of action of current point-of-care autologous therapy treatments for osteoarthritis of the knee” is an original review article which aim to summarize autologous therapies for the treatment of osteoarthritis of the knee.

On my opinion the article is interesting, well written, with good English.

However, I highlighted some issues.

  • English language: Minor spell check is required.
  • Title: I would consider including at the end of the title this sentence "a narrative review".
  • Abstract: Please structure the abstract to attract the reader's attention,and to adapt it accordingly (purpose of the revision etc.).
  • Introduction: This section has been properly prepared.Below I would insert a section "Materials and methods" in which to indicate the databases and the methods of search and selection of articles (for example PRISMA [www.prisma-statement.org]) and relative “Results” section.
  • Section 2: Please improve this section. I would suggest titling the section "Platelet concentrates". I would suggest inserting this sentence on page 4 at line 101: <<Efficacy of platelet concentrates is still debated [PMID: 31116189]>>. Have other platelet concentrates (like platelet-rich fibrin; PRF) been studied?
  • Conclusion: Further studies will be needed to confirm the authors' hypotheses.
  • Figures: Very nice.

After making the indicated changes, I am available for a second round of peer review.

Author Response

Thank you for your careful review and consideration of our manuscript.  We hope we have addressed your concerns adequately. 

Reviewer 2

  • English language: Minor spell check is required.

The document has been spell checked.  Thank you.

  • Title: I would consider including at the end of the title this sentence "a narrative review".

Thank you for this suggestion.  The title has been altered to accommodate this request.

  • Abstract: Please structure the abstract to attract the reader's attention, and to adapt it accordingly (purpose of the revision etc.).

Thank you for this suggestion. The following statements were added to the abstract to add specificity to the topics discussed. (Lines 19-28).

This review analyzes the evidence for and against inclusion of white blood cells, the potential role of platelets, and the less studied potential role blood plasma provides when combining these components to create an autologous point of care therapy treat OA. There has been significant focus on the differences between the various autologous therapies. However, evidence suggests that there may be more in common between groups and perhaps we should be thinking of these therapies on a spectrum of the same technology, each providing significant levels of anti-inflammatory cytokines that can be antagonists against the inflammatory cytokines driving OA symptoms and progression. While clinical data has demonstrated symptom alleviation, more studies will need to be conducted to see if these pre-clinical disease-modifying findings translate into clinical practice.

  • Introduction: This section has been properly prepared. Below I would insert a section "Materials and methods" in which to indicate the databases and the methods of search and selection of articles (for example PRISMA [www.prisma-statement.org]) and relative “Results” section.

Thank you for this suggestions.  The headers for Methods and Results have been added. The following methods section was added to include methodology described in the PRISMA review. (Lines

Search terms in both PubMed database and Google Scholar included “platelet-rich plasma”, PRP, osteoarthritis, “cell culture”, preclinical, and “mechanism of action” in various combinations. The same combinations were repeated for “concentrated bone marrow aspirate”, “cBMA”, “BMAC”, “autologous anti-inflammatory”, and “AAI”. Additionally, the authors’ own knowledge and work in the field was included when relevant. A focus was made on the specific role these therapies play in the treatment of OA, and therefore cartilage specific studies were only included to expand on the potential role cartilage repair has during treatment of OA.

  • Section 2: Please improve this section. I would suggest titling the section "Platelet concentrates". I would suggest inserting this sentence on page 4 at line 101: <<Efficacy of platelet concentrates is still debated [PMID: 31116189]>>. Have other platelet concentrates (like platelet-rich fibrin; PRF) been studied?

The term platelet concentrate was added to line 88. However, this review focused on liquid formulations that could be injected intra-articularly in the treatment of OA.  As PRF is solid (gel), it is more frequently used when it can be applied directly, and often even sutured, to a defect (cartilage, bone, wound, etc.). The Giudice et al. reference was added in describing PRF in the PRP section.

(Lines 102-104):

Platelet concentrates can also be activated to form a fibrin clot, often referred to as platelet-rich fibrin (PRF). These gel products are often utilized in direct application to a surgical or wound site [8], but are not included in this OA review.

  • Conclusion: Further studies will be needed to confirm the authors' hypotheses.

This statement was added. (Line 326).

  • Figures: Very nice.

After making the indicated changes, I am available for a second round of peer review.

8. Giudice, A., et al., Dental extractions for patients on oral antiplatelet: a within-person randomised controlled trial comparing haemostatic plugs, advanced-platelet-rich fibrin (A-PRF+) plugs, leukocyte- and platelet-rich fibrin (L-PRF) plugs and suturing alone. Int J Oral Implantol (Berl), 2019. 12(1): p. 77-87.

Round 2

Reviewer 2 Report

Title. Replace semicolon with colon. 

To the authors' knowledge of PRF, there is a liquid form called injectable-PRF (i-PRF; doi:10.1111/dth.13334). 

After this change, the article will be suitable for publication.